# UNIFIED UNCERTAINTY CALIBRATION

## ABSTRACT

To build robust, fair, and safe AI systems, we would like our classifiers to say "I don't know" when facing test examples that are difficult or fall outside of the training classes. The ubiquitous strategy to predict under uncertainty is the simplistic *reject-or-classify* rule: abstain from prediction if epistemic uncertainty is high, classify otherwise. Unfortunately, this recipe does not allow different sources of uncertainty to communicate with each other, produces miscalibrated predictions, and it does not allow to correct for misspecifications in our uncertainty estimates. To address these three issues, we introduce *unified uncertainty calibration (U2C)*, a holistic framework to combine aleatoric and epistemic uncertainties. U2C enables a clean learning-theoretical analysis of uncertainty estimation, and outperforms reject-or-classify across a variety of ImageNet benchmarks.

## 1 INTRODUCTION

How can we build AI systems able to say "I do not know"? This is the problem of uncertainty estimation, key to building robust, fair, and safe prediction pipelines (Amodei et al., 2016). In a perspective for *Nature Machine Intelligence*, Begoli et al. (2019) defend that AI holds extraordinary promise to transform medicine, but acknowledges

> the reluctance to delegate decision making to machine intelligence in cases where patient safety is at stake. To address some of these challenges, medical AI, especially in its modern data-rich deep learning guise, needs to develop a principled and formal uncertainty quantification.

Endowing models with the ability to recognize when "they do not know" gains special importance in the presence of distribution shifts (Arjovsky et al., 2019). This is because uncertainty estimates allow predictors to abstain when facing anomalous examples beyond their comfort zone. In those situations, aligned AI's should delegate prediction—say, the operation of an otherwise self-driving vehicle—to humans.

The problem of uncertainty estimation in AI systems is multifarious and subject to a vast research program (Gawlikowski et al., 2023; Abdar et al., 2021; Ruff et al., 2021; Yang et al., 2021). Yet, we can sketch the most common approach to prediction under uncertainty in a handful of lines of code. Consider a neural network trained on $c = 2$ in-domain classes, later deployed in a test environment where examples belonging to unseen categories can spring into view, hereby denoted by the out-domain class $c + 1$. Then, the ubiquitous *reject-or-classify* (Chow, 1957; 1970, RC) recipe implements the following logic:

```
1  def reject_or_classify(f, u, x, theta=10):
2      # Compute softmax vector [0.1, 0.9], using classifier, describing aleatoric uncertainty
3      s_x = s(f(x))
4      # Does our epistemic uncertainty exceed a threshold?
5      if u(x) >= theta:
6        # yes: abstain with label c + 1 and total confidence
7        return [0, 0, 1]
8      else:
9        # no: predict in-domain with total confidence
10       return s_x + [0]
```

This code considers a softmax vector summarizing our *aleatoric* uncertainty about the two in-domain classes, together with a real-valued *epistemic* uncertainty. If our epistemic uncertainty exceeds a

certain threshold, we believe that the test input $x$ belongs to unseen out-domain category, so we abstain from prediction, hereby signaled as a third class. Else, we classify the input into one of the two in-domain categories, according to the softmax vector.

There are three problems with this recipe. First, different types of uncertainty cannot "communicate" with each other, so we may reject easy-to-classify examples or accept for prediction out-domain examples. Second, the RC process results in miscalibrated decisions, since RC abstains or predicts only with absolute (binary) confidence. Third, the recipe does not allow us to correct for any misspecification in the epistemic uncertainty estimate.

*Contribution*   To address the issues listed above, we introduce *unified uncertainty calibration* (U2C), a framework to integrate aleatoric and epistemic uncertainties into well-calibrated predictions. Our approach blends aleatoric and epistemic softly, allowing them to talk to each other. The resulting probabilistic predictions are well calibrated jointly over the $c + 1$ classes covering predictions and abstentions. Finally, our approach allows non-linear calibration of epistemic uncertainty, resulting in an opportunity to correct for misspecifications or reject in-domain examples. Our framework allows a clean theoretical analysis of uncertainty estimation, and yields state-of-the-art performance across a variety of standard ImageNet benchmarks. Our code is publicly available at:

https://github.com/anonymous

Our exposition is organized as follows. Section 2 reviews the basic supervised learning setup, setting out the necessary notations. Section 3 surveys current trends to estimate different types of uncertainty, namely aleatoric (Subsection 3.1) and epistemic (Subsection 3.2). In Subsection 3.3 we explain the commonly used reject-or-classify recipe to combine different sources of uncertainty, and raise some concerns about this practice. Section 4 introduces *unified uncertainty calibration*, an unified framework addressing our concerns. Section 5 provides some theoretical results about its behavior, and Section 6 evaluates the efficacy of our ideas across a variety of standard benchmarks. Finally, we close our discussion in Section 7 with some pointers for future work.

*Related work*   A body of work has looked into developing better uncertainty estimators, both aleatoric or epistemic. Our goal is to combine these two kinds of estimators efficiently. Also related to us is a recent line of work that measures uncertainty under distribution shift (Wald et al., 2021; Yu et al., 2022; Tibshirani et al., 2019); unlike us, they assume access to out-domain data, either real or artificially generated through augmentations (Lang et al., 2022).

## 2   LEARNING SETUP

Our goal is to learn a classifier $f$ mapping an input $x_i \in \mathbb{R}^d$ into its label $y_i \in \{1, \ldots c\}$. In the sequel, each $x_i$ is an image displaying one of $c$ possible objects. We consider neural network classifiers of the form $f(x_i) = w(\phi(x_i))$, where $\phi(x_i) \in \mathbb{R}^{d'}$ is the representation of $x_i$. The classifier outputs logit vectors $f(x_i) \in \mathbb{R}^c$, where $f(x_i)_j$ is a real-valued score proportional to the log-likelihood of $x_i$ belonging to class $j$, for all $i = 1, \ldots, n$ and $j = 1, \ldots, c$. Let $s$ be the softmax operation normalizing a logit vector $f(x_i)$ into the probability vector $s(f(x_i))$, with coordinates

$$s(f(x_i))_j = s_f(x_i)_j = \frac{\exp(f(x_i))_j}{\sum_{k=1}^{c} \exp(f(x_i))_k},$$

for all $i = 1, \ldots, n$, and $j = 1, \ldots, c$. Bearing with us for two more definitions, let $h_f(x_i) = \text{argmax}_{j \in \{1, \ldots, j\}} f(x_i)_j$ to be the hard prediction on $x_i$, where $h_f(x_i) \in \{1, \ldots, c\}$. Analogously, define $\pi_f(x_i) = \max_{c \in \{1, \ldots, c\}} s(f(x_i))_j$, as the prediction confidence on $x_i$, where $s(\cdot)$ ensures that $\pi_f(x_i) \in [0, 1]$.

To train our deep neural network, we access a dataset $\mathcal{D} = \{(x_i, y_i)\}_{i=1}^n$ containing *in-domain* examples $(x_i, y_i)$ drawn iid from the probability distribution $P^{\text{in}}(X, Y)$, and we search for the empirical risk minimizer (Vapnik, 1998):

$$f = \underset{\tilde{f}}{\text{argmin}} \frac{1}{n} \sum_{i=1}^{n} \ell(\tilde{f}(x_i), y_i).$$

Once trained, our classifier faces new inputs $x'$ from the *extended test distribution*, which comprises of a mixture of test inputs drawn from the in-domain distribution $P^{\text{in}}(X, Y)$ and inputs drawn from an out-of-domain distribution $P^{\text{out}}(X, Y)$. The out-domain test examples do not belong to any of the $c$ classes observed during training—we formalize this by setting $y = c + 1$ for every test example $(x, y)$ drawn from $P^{\text{out}}(X, Y)$.

Central to our learning setup is that we *do not observe* any out-domain data during training. During testing, the machine observes a mixture of in-domain and out-domain data, with no supervision as to what is what. To address out-domain data, we extend our neural network $f$ as $f^\star$, now able to predict about $x$ over $c + 1$ classes, with corresponding hard labels $h_{f^\star}(x)$ and predictive confidences $\pi_{f^\star}(x)$.

Under the test regime described above, we evaluate the performance of the procedure $f^\star$ by means of two metrics. On the one hand, we measure the average classification error

$$\text{err}_P(f^\star) = \Pr_{(x,y)\sim P}\left[h_{f^\star}(x) \neq y\right]. \tag{1}$$

On the other hand, to evaluate our estimate of confidence, we look at the expected calibration error

$$\text{ece}_P(f^\star) = \mathbb{E}_{(x,y)\sim P}\mathbb{E}_{p\sim U[0,1]}\left[\left|\Pr\left(h_{f^\star}(x) = y \mid \pi_{f^\star}(x) = p\right) - p\right|\right]. \tag{2}$$

Roughly speaking, neural networks with small ece produce calibrated confidence scores, meaning $\pi_{f^\star}(x_i) \approx P(Y = y_i \mid X = x_i)$. As a complement to ece, we look at the expected negative log-likelihood

$$\text{nll}_P(f^\star) = \mathbb{E}_{(x,y)\sim P}\left[-\log(\pi_{f^\star}(x))_y\right]. \tag{3}$$

## 3 Current trends for uncertainty estimation

Most literature differentiates between aleatoric and epistemic uncertainty (Kendall and Gal, 2017; Der Kiureghian and Ditlevsen, 2009; Hüllermeier and Waegeman, 2021). In broad strokes, we consider two sources of uncertainty by factorizing the density value of a training example $(x, y)$ drawn from the in-domain distribution $P^{\text{in}}(X, Y)$:

$$p^{\text{in}}(x, y) = \underbrace{p^{\text{in}}(y \mid x)}_{\text{aleatoric}} \cdot \underbrace{p^{\text{in}}(x)}_{\text{epistemic}}. \tag{4}$$

As implied above, (i) aleatoric uncertainty concerns the irreducible noise inherent in annotating each input $x$ with its corresponding label $y$, and (ii) epistemic uncertainty relates to the atypicality of the input $x$. When learning from a dataset containing images of cows and camels, a good predictor raises its aleatoric uncertainty when a test image does depict a cow or a camel, yet it is too blurry to make a decision; epistemic uncertainty fires when the image depicts something other than these two animals—for instance, a screwdriver. We review these statistics, as well as prominent algorithms to estimate them from data, in Subsections 3.1 and 3.2.

Given estimates for aleatoric and epistemic uncertainty, one needs a mechanism to combine this information into a final decision for each test input: either classify it into one of the $c$ in-domain categories, or abstain from prediction. We review in Subsection 3.3 the most popular blend, known as *reject-or-classify* (Chow, 1957; 1970). Here, the machine decides whether to classify or abstain by looking at the epistemic uncertainty estimate in isolation. Then, if the epistemic uncertainty estimate exceeds a threshold, the machine abstains from predictor. Else, the example the machine classifies the input into one of the $c$ in-domain categories. As we will discuss in the sequel, the reject-or-classify has several problems, that we will address with our novel framework of unified calibration.

### 3.1 Estimation of aleatoric uncertainty

The word *aleatoric* has its roots in the Latin *āleātōrius*, concerning dice-players and their games of chance. In machine learning research, aleatoric uncertainty arises due to irreducible sources of randomness in the process of labeling data. Formally, the aleatoric uncertainty of an example $(x, y)$ is a supervised quantity, and relates to the conditional probability $P(Y = y \mid X = x)$. If the true data generation process is such that $P(Y = y \mid X = x) = 0.7$, there is no amount of additional data that we could collect in order to reduce our aleatoric uncertainty about $(x, y)$—it is irreducible and intrinsic to the learning task at hand.

In practice, a classifier models aleatoric uncertainty if it is well calibrated (Guo et al., 2017; Wang et al., 2021), namely it satisfies $\pi_f(x) \approx P(Y = y \mid X = x)$ for all examples $(x, y)$. In a well-calibrated classifier, we can interpret the maximum softmax score $\pi_f(x)$ as the probability of the classifier assigning the right class label to the input $x$. However, modern machine learning models are not well calibrated by default, often producing over-confident predictions.

A common technique to calibrate deep neural network classifiers is Platt scaling (Platt, 1999). The idea here is to draw a fresh validation set $\{(x_i^{\text{va}}, y_i^{\text{va}})\}_{i=1}^{m}$ drawn from the in-domain distribution $P^{\text{in}}(X, Y)$, and optimize the cross-entropy loss to find a real valued *temperature* parameter $\tau$ to scale the logits. Given $\tau$, we deploy the calibrated neural network $f_\tau(x) = f(x)/\tau$. Guo et al. (2017) shows that Platt scaling is an effective tool to minimize the otherwise non-differentiable metric of interest, ece (2). However calibrated, such classifier lacks a mechanism to determine when a test input does not belong to any of the $c$ classes described by the in-domain distribution $P^{\text{in}}(X, Y)$. Such mechanisms are under the purview of epistemic uncertainty, described next.

## 3.2 ESTIMATION OF EPISTEMIC UNCERTAINTY

From Ancient Greek ἐπιστήμη, the word *epistemic* relates to the nature and acquisition of knowledge. In machine learning, we can relate the epistemic uncertainty $u(x)$ of a test input $x$ to its in-domain input density $p^{\text{in}}(X = x)$: test inputs with large in-domain density values have low epistemic uncertainty, and vice-versa. In contrast to aleatoric uncertainty, we can reduce our epistemic uncertainty about $x$ by actively collecting new training data around $x$. Therefore, our epistemic uncertainty is not due to irreducible randomness, but due to lack of knowledge—What is $x$ like?—or *episteme*.

Epistemic uncertainty is an unsupervised quantity, and as such it is more challenging to estimate than its supervised counterpart, aleatoric uncertainty. In practical applications, it is not necessary—nor feasible—to estimate the entire in-domain input density $p^{\text{in}}(X)$, and simpler estimates suffice. The literature has produced a wealth of epistemic uncertainty estimates $u(x)$, reviewed in surveys (Gawlikowski et al., 2023; Abdar et al., 2021; Ruff et al., 2021; Yang et al., 2021) and evaluated across rigorous empirical studies (Nado et al., 2021; Belghazi and Lopez-Paz, 2021; Ovadia et al., 2019; Yang et al., 2022). We recommend the work of Yang et al. (2022) for a modern comparison of a multitude of uncertainty estimates. For completeness, we list some examples below.

- The negative maximum logit (Hendrycks et al., 2019a, MaxLogit) estimates epistemic uncertainty as $u(x_i) = -\max_j f(x_i)_j$. Test inputs producing large maximum logits are deemed certain, and vice-versa.

- Feature activation reshaping methods set to zero the majority of the entries in the representation space $\phi(x)$. The competitive method ASH (Djurisic et al., 2022) sets to a constant the surviving entries, resulting in a sparse and binary representation space.

- Methods based on Mahalanobis distances (Lee et al., 2018; Van Amersfoort et al., 2020; Ren et al., 2021, Mahalanobis) estimate one Gaussian distribution per class in representation space. Then, epistemic uncertainty is the Mahalanobis distance between the test input and the closest class mean.

- $k$-nearest neighbor approaches (Sun et al., 2022, KNN) are a well-performing family of methods. These estimate epistemic uncertainty as the average Euclidean distance in representation space between the test input and the $k$ closest inputs from a validation set.

- Ensemble methods, such as deep ensembles (Lakshminarayanan et al., 2017), multiple-input multiple-output networks (Havasi et al., 2020), and DropOut uncertainty (Gal and Ghahramani, 2016, Dropout) train or evaluate multiple neural networks on the same test input. Then, epistemic uncertainty relates to the variance across predictions.

Choosing the right epistemic uncertainty estimate $u(x)$ depends on multiple factors, such as the preferred inductive bias, as well as our training and testing budgets. For example, the logit method requires no compute in addition to $f(x)$, but often leads to increasing epistemic *certainty* as we move far away from the training data (Hein et al., 2019). In contrast, local methods are not vulnerable to this "blindness with respect overshooting", but require more computation at test time—see Mahalanobis methods—or the storage of a validation set in memory, as it happens with $k$NN methods. Finally, there is power in building our uncertainty estimate $u(x)$ from scratch (Hendrycks et al., 2019b), instead of implementing it on top of the representation space of our trained neural network. This is

because neural network classifiers suffer from a simplicity bias (Shah et al., 2020) that removes the information about $x$ irrelevant to the categorization task at hand. But, this information may be useful to signal high epistemic uncertainty far away from the training data.

For the purposes of this work, we consider $u(x)$ as given, and focus our efforts on its integration with the $c$-dimensional in-domain logit vector. Our goal is to produce a meaningful $(c + 1)$-dimensional probability vector leading to good classification error and calibration over the extended test distribution. This is an open problem as of today, since aleatoric and epsistemic uncertainty estimates combine in a simplistic manner, the reject-or-classify recipe.

### 3.3 Reject or classify: simplistic combination of uncertainties

We are now equipped with a calibrated neural network $f_\tau$—able to discern between $c$ in-domain classes—and an epistemic uncertainty estimator $u$—helpful to determine situations where we are dealing with anomalous or out-of-distribution test inputs. The central question of this work emerges: when facing a test input $x$, how should we combine the information provided by the $c$ real-valued scores in $f_\tau(x)$ and the real-valued score $u(x)$, as to provide a final probabilistic prediction?

Prior work implements a *reject or classify* (Chow, 1957; 1970, RC) recipe. In particular, classify test input $x$ as

$$\hat{y} = \begin{cases} h_{f_\tau}(x) & \text{if } u(x) < \theta, \\ c + 1 & \text{else.} \end{cases} \tag{5}$$

In words, RC classifies as out-of-distribution (with a label $\hat{y} = c+1$) those examples whose epistemic uncertainty exceeds a threshold $\theta$, and assigns an in-domain label ($\hat{y} = \hat{c} \in \{1, \ldots, c\}$) to the rest. Common practice employs a fresh validation set $\{(x_i^{\text{va}})\}_{i=1}^m$ drawn iid from the in-domain distribution $P^{\text{in}}(X, Y)$ to compute the threshold $\theta$. One common choice is to set $\theta$ to the $\alpha = 0.95$ percentile of $u(x^{\text{va}})$ across the validation inputs. This results in "giving-up" classification on the $5\%$ most uncertain inputs from the in-domain distribution—and all of those beyond—according to the epistemic uncertainty measure $u$.

Overall, we can express the resulting RC pipeline as a machine producing *extended* $(c + 1)$-dimensional softmax vectors

$$s_{\text{RC}}^\star(x) = \text{concat}\left(s\left(f_\tau(x)_1, \ldots, f_\tau(x)_c\right) \cdot [u(x) < \theta], 1 \cdot [u(x) \geq \theta]\right). \tag{6}$$

We argue that this construction has three problems. First, aleatoric and epistemic uncertainties do not "communicate" with each other. In the common cases where $u(x)$ is misspecified, we may reject in-domain inputs that are easy to classify, or insist on classifying out-domain inputs. Second, the softmax vector (6) is not calibrated over the extended problem on $c + 1$ classes, as we always accept and reject with total confidence, resulting in a binary $(c + 1)$-th softmax score. Third, the uncertainty estimate $u(x)$ may speak in different units than the first $c$ logits. To give an example, it could happen that $u(x)$ grows too slowly as to "impose itself" on out-domain examples against the in-domain logits.

## 4 Unified Uncertainty calibration: a holistic approach

To address the problems described above, we take a holistic approach to uncertainty estimation by learning a good combination of aleatoric and epistemic uncertainty. Formally, our goal is to construct an extended softmax vector, over $c + 1$ classes, resulting in low test classification error and high calibration jointly over in-domain and out-domain data. Our approach, called *unified uncertainty calibration* (U2C), works as follows. First, collect a fresh validation set $\{(x_i^{\text{va}}, y_i^{\text{va}})\}_{i=1}^m$ from the in-domain distribution $P^{\text{in}}(X, Y)$. Second, compute the threshold $\theta$ as the $\alpha = 0.95$ percentile of $u(x_i^{\text{va}})$ across all inputs in the validation set. Third, relabel those $5\%$ examples with $y_i^{\text{va}} = c + 1$. Finally, learn a non-linear epistemic calibration function $\tau_u : \mathbb{R} \to \mathbb{R}$ by minimizing the cross-entropy on the relabeled validation set:

$$\tau_u = \underset{\tilde{\tau}_u}{\arg\min} - \sum_{i=1}^m \log \text{concat}\left(f_\tau(x_i^{\text{va}}), \tilde{\tau}_u(x_i^{\text{va}})\right)_{y_i^{\text{va}}}. \tag{7}$$

After finding $\tau_u$, our U2C pipeline deploys a machine producing *extended* $(c + 1)$-dimensional softmax vectors:

$$s^\star_{\text{U2C}}(x) = s\left(f_\tau(x)_1, \ldots f_\tau(x)_c, \tau^u(u(x))\right) \tag{8}$$

The construction (8) has three advantages, addressing the three shortcomings from the the previous RC (6). First, aleatoric and epistemic uncertainties now communicate with each other by sharing the unit norm of the produced extended softmax vectors. Because (8) can describe both aleatoric and epistemic uncertainty non-exclusively, there is the potential to identify easy-to-classify examples that would otherwise be rejected. Second, we can now calibrate the extended softmax vectors (8) across the extended classification problem of $c + 1$ classes. For instance, we could now reject examples with different levels of confidence. Third, the *non-linear* epistemic calibration $\tau_u(u(x))$ has the potential to allow all of the logits to "speak the same units", such that aleatoric and epistemic uncertainty have appropriate rates of growth.

Unified calibration reduces the difficult problem of combining aleatoric and epistemic uncertainty over $c$ classes into the easier problem of optimizing for aleatoric uncertainty over $c + 1$ classes. This allows us to use (nonlinear!) Platt scaling to optimize the ece over the extended problem. In addition, the extended softmax vectors provided by U2C allow reasoning in analogy to the well-known *quadrant of knowledge* (Monarch, 2019). To see this, consider a binary classification problem with uncertainties jointly calibrated with U2C, resulting in three-dimensional extended softmax vectors that describe the probability of the first class, second class, and out-domain class. Then,

- vectors such as $(0.9, 0.1, 0.0)$ are *known-knowns*, things that we are aware of (we can classify) and we understand (we know how to classify), no uncertainty;

- vectors such as $(0.4, 0.6, 0.0)$ are *known-unknowns*, things we are aware of but we do not understand. These are instances with aleatoric uncertainty, but no epistemic uncertainty;

- vectors such as $(0.1, 0.0, 0.9)$ are *unknown-knowns*, things we understand but are not aware of. These are instances with epistemic uncertainty, but no aleatoric uncertainty.

Finally, there are *unknown-unknowns*, things that we are not aware of, nor understand. These are patterns not included in the current representation space—as such, we say that the model is "myopic" with respect those features (Belghazi, 2024). Unknown-unknowns are a necessary evil when learning about a complex world with limited computational resources (Vervaeke et al., 2012). Otherwise, any learning system would have to be aware of a combinatorially-explosive amount of patterns to take the tiniest decision—a paralyzing prospect. Rather cruel experiments with cats (Blakemore and Cooper, 1970) show how unknown-unknowns relate to biological learning systems: kittens housed from birth in an environment containing only vertical stripes were, later in life, unable to react to horizontal stripes.

## 5    THEORETICAL RESULTS

We attempt to understand the relative performance of RC and U2C by looking closely at where data points from $P^{\text{in}}$ and $P^{\text{out}}$ lie. Observe that reject-or-classify rejects when $u(x) \geq \theta$, and unified uncertainty calibration rejects when $\max_i f_\tau(x)_i \leq \tau(u(x))$; to understand their relative differences, we look at the space induced by $\tau(u(x))$ and the max-logit.

Figure 1a shows that the accept/reject regions break up the space into four parts: $A$, where both methods predict with the neural network $f$, $B$, where U2C rejects but not RC, $C$, where RC rejects but not U2C, and $D$, where both reject. $A$ is a clear in-distribution region of high confidence predictions, and $D$ is a clear out-of-distribution zone with high uncertainty. More interesting are the regions $B$ and $C$; in $C$, the uncertainty is high but max-logits are higher. This is the "Dunning-Kruger" region—little data is seen here during training, yet the network is highly confident. Conversely, $B$ is the region of high in-distribution aleatoric uncertainty, with low to moderate epistemic uncertainty.

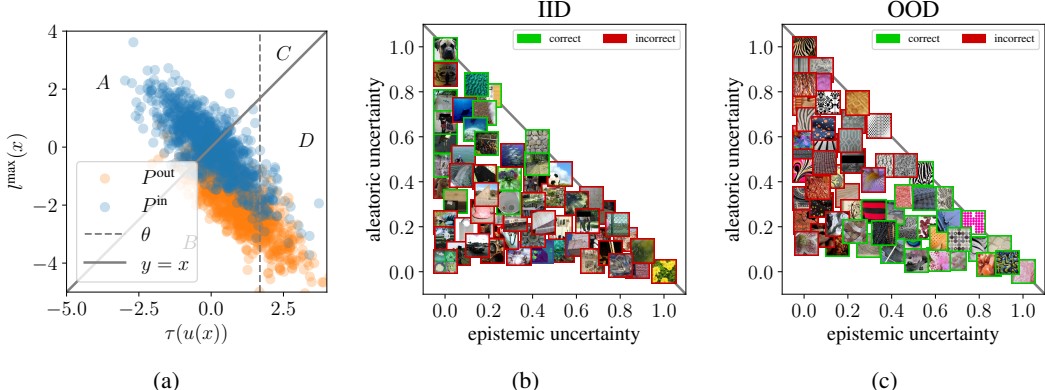

(a)  (b)  (c)

Figure 1: Panel (a) shows the acceptance/rejection regions of RC and U2C, serving as a visual support to our theoretical analysis. Panel (b) shows examples of IID images according to their epistemic uncertainty ($u(x)$, horizontal axis), aleatoric uncertainty ($\pi_f(x)$, vertical axis), and correctness of classification (border color). Panel (c) illustrates OOD images similarly. Last two panels illustrate how U2C covers all possible aleatoric-epistemic combinations, in way that correlates appropriately to (mis)classification, both IID and OOD.

**Lemma 5.1.** *The difference of errors between RC and U2C based on a network $f_\tau$ is:*

$$err_{P^{out}}(RC) - err_{P^{out}}(U2C) = P^{out}(B) - P^{out}(C)$$
$$err_{P^{in}}(RC) - err_{P^{in}}(U2C) = P^{in}(C) - P^{in}(B)$$
$$+ P^{in}(B) \cdot err_{P^{in}}(h_{f_\tau}(x)|x \in B)$$
$$- P^{in}(C) \cdot err_{P^{in}}(h_{f_\tau}(x)|x \in C).$$

If $P^{out}$ has a lot of mass in $B$, and little in $C$, then U2C outperforms RC. $B$ is the region of high aleatoric uncertainty and low to moderate epistemic uncertainty, and hence communication between different kinds of uncertainties helps improve performance. In contrast, if $P^{in}$ has a lot of mass in $C$ but little in $B$, then RC outperforms U2C in terms of hard predictions. The training loss for $\tau$ ensures that at least $95\%$ of the validation data lies in $A$ and at most $5\%$ in $D$. Therefore, if the underlying neural network has high accuracy, and if $\tau$ generalizes well, then we expect $P^{in}(B \cup C)$ to be low.

A related question is what happens in $C$, which is the region where U2C predicts with high confidence yet low evidence. Since both the max-logit and the uncertainty are complex functions of $x$, all possible values of $(\max_i(f_\tau(x))_i, \tau(u(x)))$ are not achievable, and varying $x$ within the instance space induce pairs within an allowable set. Choosing $u$ to limit that allowable set will permit us to bound $C$. For example, for binary linear classification, if we ensure that the uncertainty estimate $u$ grows faster than the logits, then $C$ will be bounded by design.

While Lemma 5.1 above analyzes hard predictions, we expect most of the advantages of U2C to be due to its ability to "softly" adjust its confidence. To understand this, Lemma 5.2 analyzes the negative log-likelihood of both methods. Analogous results for ece are in the Appendix.

**Lemma 5.2.** *The nll of U2C based on a network $f_\tau$ is given by:*

$$nll_{P^{out}}(U2C) = -\mathbb{E}_{x \sim P^{out}}\left[\log \frac{e^{\tau(u(x))}}{\sum_{j=1}^c e^{f_\tau(x)_j} + e^{\tau(u(x))}}\right], \quad x \sim P^{out}.$$

$$nll_{P^{in}}(U2C) = -\mathbb{E}_{(x,y) \sim P^{in}}\left[\log \frac{e^{f_\tau(x)_y}}{\sum_{j=1}^c e^{f_\tau(x)_j} + e^{\tau(u(x))}}\right], \quad x \sim P^{in}.$$

*If $x \sim P^{out}$, then the nll of RC is $0$ for $x \in C \cup D$, and $\infty$ for $x \in A \cup B$. If $x \sim P^{in}$, the nll of RC is as follows:*

$$nll_{P^{in}}(RC) = \infty, \quad x \in C \cup D.$$
$$= P^{in}(A \cup B) \cdot \mathbb{E}_{(x,y) \sim P^{in}}[nll(f_\tau(x))|x \in A \cup B], \quad x \in A \cup B.$$

Lemma 5.2 implies that the nll of RC will be infinite if $P^{\text{in}}$ has some probability mass in $C \cup D$; this is bound to happen since the construction of $\tau$ ensures that $5\%$ of in-distribution examples from $P^{\text{in}}$ are constrained to be in $D$. The negative log-likelihood of RC will also be infinite if $P^{\text{out}}$ has some probability mass in $A \cup B$, which is also likely to happen. This is a consequence of the highly confident predictions made by RC. In contrast, U2C makes softer predictions that lower nll values.

## 6  EXPERIMENTS

We now turn to the empirical comparison between RC and U2C. Our main objective is to show that unified uncertainty calibration achieves better performance metrics, namely err (1) and ece (2), over both the in-domain and out-domain data.

*Benchmarks*    We perform a full-spectrum out-of-distribution detection analysis (Zhang et al., 2023), evaluating metrics on four types of ImageNet benchmarks: in-domain, covariate shift, near-ood, and far-ood. First, to evaluate in-domain we construct two equally-sized splits of the original ImageNet validation set (Deng et al., 2009), that we call ImageNet-va and ImageNet-te. Our split ImageNet-va is used to find the epistemic uncertainty threshold $\theta$ and calibration parameters $(\tau, \tau_u)$. The split ImageNet-te is our true in-domain "test set", and models do not have access to it until evaluation. Second, we evaluate metrics under covariate shift using the in-domain datasets ImageNet-C (Hendrycks and Dietterich, 2019) containing image corruptions, ImageNet-R (Hendrycks et al., 2021a) containing artistic renditions, and the ImageNet-v2 validation set (Recht et al., 2019). For these first two benchmarks, we expect predictors to classify examples $x$ into the appropriate in-domain label $y \in \{1, \ldots, c\}$. Third, we evaluate metrics for the near-ood datasets NINCO (Bitterwolf et al., 2023) and SSB-Hard (Vaze et al., 2021). Near-ood datasets are difficult out-of-distribution detection benchmarks, since they contain only examples from the out-of-distribution class $y = c + 1$, but are visually similar to the in-domain classes. Finally, we evaluate metrics on the far-ood datasets iNaturalist (Huang and Li, 2021), Texture (Cimpoi et al., 2014), and OpenImage-O (Wang et al., 2022). Far-ood datasets also contain only examples from the out-of-distribution class $y = c + 1$, but should be easier to distinguish from those belonging to the in-domain classes.

*Epistemic uncertainty estimates*    Both methods under comparison, RC and U2C, require the prescription of an epistemic uncertainty estimate $u(x)$. We turn to the fantastic survey OpenOOD (Zhang et al., 2023) and choose four high-performing alternatives spannign different families. These are the MaxLogit (Hendrycks et al., 2019a), ASH (Djurisic et al., 2022), Mahalanobis (Ren et al., 2021), and KNN (Sun et al., 2022) epistemic uncertainty estimates, all described in Section 3.2.

*Results*    Table 1 shows err/ece metrics for RC, for all the considered benchmarks and epistemic uncertainty estimates. In parenthesis, we show the improvements (in green) or deteriorations (in red) brought about by replacing RC by our proposed U2C. Error-bars are absent because there is no randomness involved in our experimental protocol—the splits ImageNet-va and ImageNet-te were computed once and set in stone for all runs. As we can see, U2C brings improvements in both test classification accuracy and calibration error in most experiments. When U2C deteriorates results, it does so with a small effect. Figures 1b and 1c shows the calibrated epistemic-aleatoric uncertainty space, covering the entire lower-triangle of values—in contrast, RC could only cover two crowded vertical bars at the two extremes of epistemic uncertainty. Appendix A.1 shows additional experiments on *linear* U2C (showcasing the importance of calibrating nonlinearly), as well as other neural network architectures, such as ResNet152 and ViT-32-B.

## 7  DISCUSSION

We close our exposition by offering some food for thought. First, the problem of unknown-unknowns (feature myopia) remains a major challenge. If color is irrelevant to a shape classification problem, should we be uncertain on how to classify known shapes of unseen colors? If so, general-purpose representations from self-supervised learning may help. However, as famously put by Goodman (1972), no representation can be aware of the combinatorially-explosive amount of ways in which two examples may be similar or dissimilar. Therefore, we will always remain blind to most features, suggesting the impossibility of uncertainty estimation without strong assumptions. Also related is the

|  |  | **MaxLogit** | **ASH** | **Mahalanobis** | **KNN** |
|---|---|---|---|---|---|
| ImageNet-va | err | 25.1 (+0.2) | 25.6 (−0.0) | 24.9 (−0.0) | 26.7 (−0.2) |
|  | ece | 7.0 (−0.7) | 7.1 (−0.6) | 7.7 (−0.6) | 7.2 (−0.9) |
| ImageNet-te | err | 25.2 (+0.2) | 25.8 (−0.0) | 34.1 (−0.5) | 27.4 (−0.3) |
|  | ece | 6.2 (−0.6) | 6.6 (−0.6) | 21.4 (−1.6) | 7.3 (−0.8) |
| ImageNet-v2 | err | 38.7 (+0.4) | 39.0 (+0.2) | 49.8 (−0.5) | 40.3 (−0.0) |
|  | ece | 14.5 (−0.1) | 13.5 (−0.2) | 35.9 (−1.5) | 12.0 (−0.0) |
| ImageNet-C | err | 67.7 (+0.5) | 69.7 (+0.2) | 77.1 (+0.2) | 72.7 (+1.0) |
|  | ece | 48.0 (−0.4) | 52.2 (−0.2) | 67.4 (−0.2) | 55.0 (+1.6) |
| ImageNet-R | err | 79.8 (+0.4) | 78.7 (+0.3) | 87.4 (−0.0) | 81.4 (+0.7) |
|  | ece | 56.3 (−1.0) | 53.1 (−0.0) | 74.9 (−0.0) | 54.5 (+2.9) |
| NINCO | err | 77.2 (−2.2) | 67.6 (−1.4) | 30.8 (−0.4) | 73.3 (−5.1) |
|  | ece | 40.3 (−3.3) | 35.4 (−2.4) | 18.6 (−1.5) | 35.1 (−4.1) |
| SSB-Hard | err | 84.8 (−1.7) | 83.2 (−1.1) | 47.2 (−0.0) | 87.1 (−2.0) |
|  | ece | 51.8 (−2.4) | 50.3 (−1.6) | 33.1 (−0.9) | 49.9 (−1.7) |
| iNaturalist | err | 51.8 (−3.5) | 15.9 (−0.2) | 16.5 (−2.2) | 58.5 (−7.4) |
|  | ece | 22.6 (−5.3) | 8.9 (−1.3) | 7.3 (−2.0) | 19.6 (−5.0) |
| Texture | err | 52.9 (−2.9) | 16.3 (+0.3) | 28.0 (−3.1) | 10.5 (−1.2) |
|  | ece | 29.8 (−4.1) | 11.1 (−0.7) | 14.6 (−2.7) | 6.0 (−1.2) |
| OpenImage-O | err | 58.6 (−3.3) | 34.6 (−1.3) | 21.5 (−1.9) | 55.3 (−5.9) |
|  | ece | 28.6 (−5.0) | 17.5 (−2.4) | 11.1 (−2.0) | 21.9 (−4.4) |

Table 1: Classification errors (err) and expected calibration errors (ece) for reject-or-classify across a variety of benchmarks and uncertainty estimates. In parenthesis, we show the metric improvements (in green) or deteriorations (in red) from using unified uncertainty calibration. Row color indicates the type of benchmark: ☐ training distribution, ☐ in-domain covariate shift, ☐ near out-of-distribution, ☐ far out-of-distribution.

issue of adversarial examples—for any trained machine, adversarial examples target exactly those features that the machine is blind to! Therefore, it is likely that adversarial examples will always exist (Hendrycks et al., 2021b).

Second, the relabeling and non-linear calibration processes in the proposed U2C are more flexible than the simple thresholding step in RC. In applications where abstention is less hazardous than misclassifying, could it be beneficial to explicitly relabel confident in-domain mistakes in the validation set as $y = c + 1$?

Third, commonly-used deep ReLU networks are famous for becoming overly confident as we move far away from the training data. Should we redesign cross-entropy losses to avoid extreme logits? Some alternatives to tame the confidence of neural networks include gradient starvation (Pezeshki et al., 2021), logit normalization (Wei et al., 2022), and mixing or smoothing labels (Zhang et al., 2017). Should we redefine $u(x) := u(x)/\|x\|$? Can we design simple unit tests for epistemic uncertainty estimates?

Looking forward, we would like to investigate the prospects that large language models (OpenAI, 2023, LLMs) bring to our discussion about uncertainty. What does framing the learning problem as next-token prediction, and the emerging capability to learn in-context, signify for the problem of estimating uncertainty? Can we aggregate uncertainty token-by-token, over the prediction of a sequence, as to guide the machine away from hallucination and other violations of factuality?

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

# A APPENDIX

## A.1 ADDITIONAL EXPERIMENTAL RESULTS

|  |  | **MaxLogit** | **ASH** | **Mahalanobis** | **KNN** |
|---|---|---|---|---|---|
| ImageNet-va | err | 25.1 (+0.2) | 25.6 (−0.3) | 24.9 (+0.3) | 26.7 (−0.7) |
|  | ece | 7.0 (−5.1) | 7.1 (−5.4) | 7.7 (−2.2) | 7.2 (−4.7) |
| ImageNet-te | err | 25.2 (+0.3) | 25.8 (−0.3) | 34.1 (−2.4) | 27.4 (−1.0) |
|  | ece | 6.2 (−4.5) | 6.6 (−5.1) | 21.4 (−7.0) | 7.3 (−4.5) |
| ImageNet-v2 | err | 38.7 (+0.5) | 39.0 (+0.2) | 49.8 (−3.0) | 40.3 (−0.2) |
|  | ece | 14.5 (−7.6) | 13.5 (−6.4) | 35.9 (−8.6) | 12.0 (−2.4) |
| ImageNet-C | err | 67.7 (+0.6) | 69.7 (−1.7) | 77.1 (−0.5) | 72.7 (−1.4) |
|  | ece | 48.0 (−26.2) | 52.2 (−29.8) | 67.4 (−3.6) | 55.0 (−15.2) |
| ImageNet-R | err | 79.8 (+0.5) | 78.7 (+1.5) | 87.4 (−0.5) | 81.4 (+0.7) |
|  | ece | 56.3 (−30.1) | 53.1 (−26.3) | 74.9 (−4.0) | 54.5 (−13.5) |
| NINCO | err | 77.2 (−2.7) | 67.6 (+7.7) | 30.8 (+1.6) | 73.3 (−9.6) |
|  | ece | 40.3 (−8.5) | 35.4 (−3.8) | 18.6 (−6.5) | 35.1 (−14.5) |
| SSB-Hard | err | 84.8 (−2.1) | 83.2 (−0.0) | 47.2 (+2.6) | 87.1 (−9.3) |
|  | ece | 51.8 (−8.4) | 50.3 (−7.2) | 33.1 (−3.8) | 49.9 (−11.6) |
| iNaturalist | err | 51.8 (−4.2) | 15.9 (+34.4) | 16.5 (−4.9) | 58.5 (−23.6) |
|  | ece | 22.6 (+9.6) | 8.9 (+20.8) | 7.3 (−6.3) | 19.6 (+0.4) |
| Texture | err | 52.9 (−3.4) | 16.3 (+35.5) | 28.0 (−7.5) | 10.5 (+16.2) |
|  | ece | 29.8 (+0.5) | 11.1 (+17.2) | 14.6 (−8.1) | 6.0 (+6.1) |
| OpenImage-O | err | 58.6 (−4.0) | 34.6 (+22.0) | 21.5 (−4.1) | 55.3 (−14.7) |
|  | ece | 28.6 (−1.8) | 17.5 (+7.4) | 11.1 (−8.0) | 21.9 (−6.6) |

Table 2: Reject-or-classify versus linear unified uncertainty calibration on ResNet50.

| | | MaxLogit | ASH | Mahalanobis | KNN |
|---|---|---|---|---|---|
| ImageNet-va | err | 23.2 (+0.2) | 23.8 (+0.1) | 23.0 (−0.0) | 24.7 (−0.2) |
| | ece | 6.6 (−0.6) | 6.8 (−0.4) | 7.2 (−0.6) | 6.8 (−0.8) |
| ImageNet-te | err | 23.0 (+0.2) | 23.6 (−0.0) | 30.7 (−0.4) | 24.9 (−0.1) |
| | ece | 6.4 (−0.5) | 6.6 (−0.5) | 19.0 (−1.4) | 7.3 (−0.6) |
| ImageNet-v2 | err | 35.2 (+0.3) | 35.4 (+0.2) | 45.6 (−0.5) | 36.7 (−0.0) |
| | ece | 13.8 (−0.3) | 12.4 (−0.2) | 32.5 (−1.4) | 11.4 (−0.0) |
| ImageNet-C | err | 62.1 (+0.4) | 64.0 (+0.3) | 70.4 (+0.1) | 65.8 (+1.0) |
| | ece | 44.6 (−0.9) | 46.9 (−0.3) | 60.5 (−0.6) | 47.3 (+1.6) |
| ImageNet-R | err | 76.5 (+0.4) | 75.2 (+0.4) | 83.2 (−0.0) | 77.8 (+0.8) |
| | ece | 56.6 (−1.3) | 51.6 (−0.0) | 73.2 (−0.3) | 56.1 (+2.4) |
| NINCO | err | 72.1 (−2.1) | 63.8 (−1.4) | 29.9 (−0.3) | 69.5 (−5.6) |
| | ece | 39.8 (−3.4) | 34.8 (−2.3) | 18.8 (−1.4) | 34.4 (−4.4) |
| SSB-Hard | err | 80.2 (−1.6) | 80.6 (−1.5) | 51.0 (−0.3) | 86.5 (−2.4) |
| | ece | 51.5 (−2.8) | 50.3 (−2.1) | 36.1 (−1.2) | 50.8 (−2.0) |
| iNaturalist | err | 46.5 (−2.8) | 15.4 (−0.3) | 15.6 (−2.2) | 47.3 (−7.4) |
| | ece | 22.3 (−4.8) | 9.1 (−1.4) | 7.4 (−1.9) | 18.4 (−5.3) |
| Texture | err | 43.4 (−2.6) | 12.1 (+0.6) | 25.4 (−3.1) | 9.7 (−1.1) |
| | ece | 25.9 (−4.3) | 8.5 (−0.3) | 14.2 (−2.6) | 6.0 (−1.0) |
| OpenImage-O | err | 51.8 (−2.6) | 31.6 (−1.2) | 19.6 (−1.9) | 49.2 (−6.0) |
| | ece | 27.0 (−4.6) | 16.8 (−2.2) | 10.7 (−2.0) | 20.2 (−4.6) |

Table 3: Reject-or-classify versus unified uncertainty calibration on ResNet152.

| | | **MaxLogit** | **ASH** | **Mahalanobis** | **KNN** |
|---|---|---|---|---|---|
| ImageNet-va | err | 20.3 (+0.1) | 22.8 (−3.6) | 19.8 (−0.0) | 20.8 (−0.0) |
| | ece | 8.4 (−0.4) | 8.5 (−5.0) | 7.4 (−0.3) | 8.1 (−0.5) |
| ImageNet-te | err | 20.3 (+0.1) | 22.9 (−3.8) | 22.1 (−0.0) | 21.1 (−0.0) |
| | ece | 8.4 (−0.4) | 8.8 (−5.4) | 11.2 (−0.5) | 8.6 (−0.6) |
| ImageNet-v2 | err | 32.3 (+0.1) | 34.0 (−3.5) | 35.2 (−0.0) | 33.1 (−0.0) |
| | ece | 18.1 (−0.7) | 14.9 (−8.6) | 23.7 (−0.9) | 17.8 (−1.1) |
| ImageNet-C | err | 50.7 (+0.4) | 46.9 (−2.2) | 52.6 (+0.1) | 49.8 (+0.6) |
| | ece | 37.6 (−1.4) | 15.2 (−8.8) | 42.5 (−1.2) | 34.9 (−1.1) |
| ImageNet-R | err | 72.9 (+0.2) | 71.7 (−1.0) | 76.4 (−0.0) | 73.8 (+0.2) |
| | ece | 59.6 (−1.7) | 34.5 (−6.5) | 67.9 (−1.6) | 62.3 (−2.4) |
| NINCO | err | 73.9 (−0.8) | 93.8 (−1.0) | 45.3 (+0.1) | 75.2 (−2.7) |
| | ece | 54.1 (−3.0) | 58.4 (−2.1) | 35.8 (−1.5) | 51.9 (−4.8) |
| SSB-Hard | err | 85.8 (−0.8) | 93.0 (+3.6) | 63.8 (+0.1) | 87.4 (−1.2) |
| | ece | 66.8 (−2.2) | 66.4 (−0.0) | 52.7 (−1.0) | 65.9 (−2.3) |
| iNaturalist | err | 52.4 (−1.4) | 91.2 (−7.1) | 12.5 (+0.3) | 55.8 (−4.5) |
| | ece | 35.5 (−4.4) | 44.2 (−0.5) | 10.6 (−0.5) | 33.4 (−7.5) |
| Texture | err | 53.8 (−1.3) | 92.2 (−9.9) | 39.9 (−1.1) | 46.3 (−3.5) |
| | ece | 39.0 (−4.0) | 48.8 (−2.0) | 29.0 (−2.8) | 30.8 (−6.5) |
| OpenImage-O | err | 58.9 (−1.6) | 91.0 (−4.2) | 28.3 (−0.3) | 59.3 (−3.6) |
| | ece | 41.3 (−4.3) | 48.7 (−1.7) | 21.9 (−1.7) | 37.0 (−6.4) |
| ImageNet-O | err | 89.1 (−0.7) | 93.8 (+3.7) | 79.4 (−0.0) | 80.0 (−0.7) |
| | ece | 75.4 (−1.8) | 74.3 (+0.7) | 68.3 (−1.6) | 67.3 (−3.4) |

Table 4: Reject-or-classify versus unified uncertainty calibration on ViT-B-16.

### A.1.1 PROOFS

*Proof.* (Of Lemma 5.1) To prove the first part, we observe that in the regions $A$ and $D$, both RC and U2C make the same hard prediction. In contrast, in $C$, RC predicts class $c + 1$ while U2C reverts to the prediction made by $f_\tau(x)$, and the converse happens in $B$. This means that when data is drawn from $P^{\text{out}}$, $\text{err}_{P^{\text{out}}}(RC) = P^{\text{out}}(A) + P^{\text{out}}(C)$, while $\text{err}_{P^{\text{out}}}(U2C) = P^{\text{out}}(A) + P^{\text{out}}(B)$. The first equation follows.

When $x$ is drawn from $P^{\text{in}}$, RC always predicts class $c + 1$ incorrectly in $C$ and $D$, and also predicts incorrectly in $A$ and $B$ if $f_\tau(x)$ is incorrect. Therefore,

$$
\begin{aligned}
\text{err}_{P^{\text{in}}}(RC) &= P^{\text{in}}(C) + P^{\text{in}}(D) + P^{\text{in}}(A) \cdot \text{err}_{P^{\text{in}}}(h_{f_\tau}(x)|x \in A) \\
&\quad + P^{\text{in}}(B) \cdot \text{err}_{P^{\text{in}}}(h_{f_\tau}(x)|x \in B) \\
\text{err}_{P^{\text{in}}}(U2C) &= P^{\text{in}}(D) + P^{\text{in}}(B) + P^{\text{in}}(A) \cdot \text{err}_{P^{\text{in}}}(h_{f_\tau}(x)|x \in A) \\
&\quad + P^{\text{in}}(C) \cdot \text{err}_{P^{\text{in}}}(h_{f_\tau}(x)|x \in C)
\end{aligned}
$$

The second equation follows. $\qquad\square$

*Proof.* (Of Lemma 5.2) To prove the first part of the lemma, we observe that when $x \sim P^{\text{out}}$, the correct label is $c + 1$, while U2C predicts the probability vector $s(f_\tau(x)_1, \ldots, f_\tau(x)_c, \tau(u(x)))$. The first two equations follow from the definition.

For RC, when $x \sim P^{\text{out}}$, the true class is $c + 1$, and the predicted probability is $(0, \ldots, 0, 1)$ in $C \cup D$, and $s(f_\tau(x)_1, \ldots, f_\tau(x)_c, 0)$ in $A \cup B$. This leads to a negative log-likelihood of zero for any $x \in C \cup D$ and infinity for any $x \in A \cup B$.

When $x \sim P^{\text{in}}$, RC predicts $(0, \ldots, 0, 1)$ in $C \cup D$ and $s(f_\tau(x)_1, \ldots, f_\tau(x)_c, 0)$, the same vector as $f_\tau$, in $A \cup B$. The last equation follows. $\qquad\square$

### A.1.2 RESULTS ON ECE

**Lemma A.1.** *If $x \sim P^{out}$, then the ece values of RC in A to D are given by:*

$$
\begin{aligned}
ece_{P^{out}|A}(RC) &= \mathbb{E}_{(x,y) \sim P^{out}|A} \left[ \max_i \frac{e^{(f_\tau(x))_i}}{\sum_{j=1}^c e^{(f_\tau(x))_j}} \right] \\
ece_{P^{out}|B}(RC) &= \mathbb{E}_{(x,y) \sim P^{out}|B} \left[ \max_i \frac{e^{(f_\tau(x))_i}}{\sum_{j=1}^c e^{(f_\tau(x))_j}} \right] \\
ece_{P^{out}|C}(RC) &= 0 \\
ece_{P^{out}|D}(RC) &= 0
\end{aligned}
$$

*Similarly, the ece values of U2C in the regions A to D are given by:*

$$
\begin{aligned}
ece_{P^{out}|A}(U2C) &= \mathbb{E}_{(x,y) \sim P^{out}|A} \left[ \max_i \frac{e^{(f_\tau(x))_i}}{\sum_{i=1}^c e^{(f_\tau(x))_i} + e^{\tau(u(x))}}, \right] \\
ece_{P^{out}|B}(U2C) &= \mathbb{E}_{(x,y) \sim P^{out}|B} \left[ \left( 1 - \frac{e^{\tau(u(x))}}{\sum_{i=1}^c e^{(f_\tau(x))_i} + e^{\tau(u(x))}} \right), \right] \\
ece_{P^{out}|C}(U2C) &= \mathbb{E}_{(x,y) \sim P^{out}|C} \left[ \max_i \frac{e^{(f_\tau(x))_i}}{\sum_{i=1}^c e^{(f_\tau(x))_i} + e^{\tau(u(x))}}, \right] \\
ece_{P^{out}|D}(U2C) &= \mathbb{E}_{(x,y) \sim P^{out}|D} \left[ \left( 1 - \frac{e^{\tau(u(x))}}{\sum_{i=1}^c e^{(f_\tau(x))_i} + e^{\tau(u(x))}} \right), \right]
\end{aligned}
$$

*Proof.* Suppose that $x \sim P^{\text{out}}$. Then, for any $x \in C \cup D$, the probability vector predicted by RC is $(0, \ldots, 0, 1)$, which is always equal to the true generating probability vector. Hence, the ece values over $C$ and $D$ are zero.

For any $x \in A$, the predicted probability vector is $s((f_\tau(x))_1, \ldots, (f_\tau(x))_c, 0)$, the maximum coordinate of which is a label in $\{1, \ldots, c\}$. At this coordinate, the predicted probability is

$\max_i \frac{e^{(f_\tau(x))_i}}{\sum_{j=1}^c e^{(f_\tau(x))_j}}$, while the actual probability is zero (since $x \sim P^{\text{out}}$). By definition of ece, this means that the ece in the region $A$ is $\int_{x \in A} \max_i \frac{e^{(f_\tau(x))_i}}{\sum_{j=1}^c e^{(f_\tau(x))_j}} P^{\text{out}}(x)dx$, from which plus conditioning, the first equation follows. A similar analysis follows for $x \in B$.

Now let us look at a similar analysis for DC. Suppose $x \in A$; then the predicted probability vector for DC is $s((f_\tau(x))_1, \ldots, (f_\tau(x))_c, \tau(u(x)))$, the maximum coordinate of which is a label in $\{1, \ldots, c\}$. At this coordinate, the predicted probability is $\max_i \frac{e^{(f_\tau(x))_i}}{\sum_{j=1}^c e^{(f_\tau(x))_j} + e^{\tau(u(x))}}$, while the actual generating probability is zero. By definition of ece, the ece of DC in A is thus $\int_{x \in A} \max_i \frac{e^{(f_\tau(x))_i}}{\sum_{j=1}^c e^{(f_\tau(x))_j} + e^{\tau(u(x))}} P^{\text{out}}(x)dx$, from which the lemma follows. A similar analysis applies to $x \in C$.

For $x \in B$, the predicted probability vector for DC is $s((f_\tau(x))_1, \ldots, (f_\tau(x))_c, \tau(u(x)))$, the maximum coordinate of which is $c + 1$. Since $x$ is drawn from $P^{\text{out}}$, for this coordinate the actual probability is always 1. Therefore, the ece will be $\int_{x \in B}(1 - \frac{e^{\tau(u(x))}}{\sum_{j=1}^c e^{(f_\tau(x))_j} + e^{\tau(u(x))}})P^{\text{out}}(x)dx$, from which the lemma follows. A similar analysis holds for $x \in D$. $\qquad\square$

**Lemma A.2.** *If $x \sim P^{\text{in}}$, then the ece values of RC in the regions A to D are given by:*

$$
\begin{aligned}
ece_{P^{\text{in}}|A}(RC) &= ece_{P^{\text{in}}|A}(f_\tau) \\
ece_{P^{\text{in}}|B}(RC) &= ece_{P^{\text{in}}|B}(f_\tau) \\
ece_{P^{\text{in}}|C}(RC) &= 1 \\
ece_{P^{\text{in}}|D}(RC) &= 1
\end{aligned}
$$

*Similarly, the ece values of U2C in B and D are:*

$$
\begin{aligned}
ece_{P^{\text{in}}|B}(U2C) &= \mathbb{E}_{(x,y)\sim P^{\text{in}}|B}\left[\frac{e^{\tau(u(x))}}{\sum_{i=1}^c e^{(f_\tau(x))_i} + e^{\tau(u(x))}}\right] \\
ece_{P^{\text{in}}|D}(U2C) &= \mathbb{E}_{(x,y)\sim P^{\text{in}}|D}\left[\frac{e^{\tau(u(x))}}{\sum_{i=1}^c e^{(f_\tau(x))_i} + e^{\tau(u(x))}}\right]
\end{aligned}
$$

*Proof.* Suppose $x \sim P^{\text{in}}$. For RC, the predicted probability vector for an $x$ in $C \cup D$ is $(0, \ldots, 0, 1)$, the maximum coordinate of which is $c + 1$. In reality, as $x$ is drawn from $P^{\text{in}}$, the actual probability of this coordinate is zero. The ece is therefore always 1.

Instead if $x \in A \cup B$, the probability vector predicted by RC is $s((f_\tau(x))_1, \ldots, (f_\tau(x))_c, 0)$, which is the same as that predicted by $h$. The ece restricted to these regions is thus the same as the ece of $h$ in these regions.

In the region $B \cup D$, DC predicts class $c + 1$ with probability vector $s((f_\tau(x))_1, \ldots, (f_\tau(x))_c, \tau(u(x)))$, while the actual probability of class $c + 1$ is zero. Therefore, the ece in $B$ is $\int_{x \in B} \frac{e^{\tau(u(x))}}{\sum_{j=1}^c e^{(f_\tau(x))_c} + e^{\tau(u(x))}} P^{\text{in}}(x)dx$, from which the lemma follows. A similar analysis holds for $x \in D$. $\qquad\square$

Observe that in $A$ and $C$, the ece of U2C is hard to analyze – since it involves a predicted probability vector that is a damped version of $f_\tau$. The ece might degrade if $f_\tau$ is perfectly or close to perfectly calibrated, or might even improve if $f_\tau$ itself is over-confident.

