# OpenReview forum: "Unified Uncertainty Estimation"
_ICLR.cc/2024/Conference — Submitted to ICLR 2024_

### Official Review · Reviewer_3np5 · 2023-10-29

**Soundness:** 3 good
**Presentation:** 3 good
**Contribution:** 3 good
**Rating:** 8
**Confidence:** 4

**Summary:**

The paper proposes a U2C framework to combine aleatoric and epistemic uncertainties in OOD classification problems. The benchmark method, reject-or-classify (RC), abstains from prediction if the epistemic uncertainty is high, and otherwise produces an in-domain prediction. The RC method follows a "hard" way to classify a sample into the out-domain class, while the proposed U2C method takes a softer approach by concatenating an epistemic uncertainty score to the logit vector of a pre-trained predictor, before passing the augmented vector to a softmax layer. The problem does not assume the accessibility of out-domain data, so the U2C method picks the most epistemically uncertain data from the training set and then replaces their original labels with a new out-domain label.

**Strengths:**

1. The idea is very natural and intuitive. The separation of epistemic uncertainty and aleatoric scores in the RC method inevitably induces various inconveniences, as discussed in the introduction, and the U2C framework is a natural attempt to construct "epistemic scores" that are comparable to aleatoric scores.

2. The U2C framework identifies a new problem formulation for uncertainty calibration, and I think one contribution of the work is to formulate this "calibration" task under the OOD setting.

3. The algorithm benefits from a good epistemic uncertainty estimator and a high-performance calibrating algorithm, making it adaptive and flexible.

**Weaknesses:**

1. The U2C algorithm treats the samples with the largest epistemic uncertainty in the training set as OOD samples. Such treatments rely on the belief that OOD samples have comparable epistemic uncertainties to the most uncertain data in the training set. What if the epistemic uncertainty completely fails to capture the uncertainty of the OOD samples? Say, for a dog-v.s.-cat binary classification task, and the OOD class is mouse, what if the trained model predicts a score of (0.8,0.2) for a mouse image that wrongly classifies the mouse as dog?

2. The theoretical part is a bit weak. For example, in Lemma 5.2 the paper criticizes the RC method by claiming that the losses pick infinity values in certain cases, while in practical implements the losses are truncated to avoid the occurrence of infinite values.

Minor comments:

1. It says "Roughly speaking, neural networks with small ece produce calibrated confidence scores, meaning XXX". This claim is wrong and misleading. Formula (2) is a "marginal" performance measure, while the XXX formula therein is an "individual"/"conditional" criterion. The marginal one can't imply an individual guarantee.

2. Sorry if I missed some details. I just wonder for the numerical experiments, do you use a pre-trained model or train the model from scratch?

**Questions:**

As above in the weakness.

---

> ### Author Response · Authors · 2023-11-14
>
> Thank you for your feedback!
>
> * You are correct that both RC and U2C will model uncertainty poorly when given the wrong $u(x)$. However, in this paper we assume $u(x)$ as given—the user can choose one of the empirically-validated state-of-the-art alternatives in the literature, such as k-nn or Mahalanobis distances.
>
> * In our experiments, we use frozen backbones pre-trained on ImageNet; we will clarify this early on in the paper.
>
> * We will clarify that small global ECE does not imply pointwise convergence to conditional probabilities.
>
> * We used infinities in the theoretical results to keep them simple and understandable; they can be replaced by expressions involving the maximum capped values times the probability masses of various regions.

---

### Official Review · Reviewer_LfhH · 2023-10-29

**Soundness:** 3 good
**Presentation:** 3 good
**Contribution:** 2 fair
**Rating:** 5
**Confidence:** 3

**Summary:**

The paper addresses the problem of uncertainty calibration in neural network classifiers. Specifically, it looks at combining aleatoric uncertainty and epistemic uncertainty. Current methods use a simple "reject-or-classify" rule to abstain from prediction when epistemic uncertainty is high. However, this approach has issues - the two uncertainties don't communicate, predictions are miscalibrated, and epistemic estimates may be misspecified.
The proposed Unified Uncertainty Calibration (U2C) framework allows both kinds of uncertainties to communicate with each other, resulting in well-calibrated probabilistic predictions. The key ideas are: relabeling the 5% most uncertain validation data as OOD. Blend aleatoric and epistemic uncertainties into an extended (c+1 class) softmax vector. Learn a non-linear calibration function of the epistemic uncertainty by minimizing the cross-entropy on the relabeled validation set. This allows the two uncertainties to communicate and produce well-calibrated probabilistic predictions.

**Strengths:**

1.	The paper introduces a novel uncertainty calibration technique (U2C) that intelligently combines aleatoric and epistemic uncertainty to improve classifier robustness. Both theoretical and empirical results demonstrate the efficacy of the proposed approach.

2.	The theoretical analysis provides useful insights into the behavior of U2C compared to the commonly used reject-or-classify recipe.

3.	The extensive experiments on ImageNet benchmarks demonstrate clear improvements from using U2C over reject-or-classify. The consistent gains across different network architectures, uncertainty estimators, and data distributions validate the efficacy of the proposed approach.

**Weaknesses:**

1.	In section 5, It seems that “In contrast, if P_in has a lot of mass in C but little in B, then RC outperforms U2C in terms of hard predictions” is incorrect. According to Figure 1 and lemma 5, it seems to be “if $P_{in}$ has more mass in C but little in B, then U2C outperforms RC.”
2.	The description of the proposed U2C framework is quite condensed. An pseudocode would help convey the steps more clearly.
3.	There are some writing errors, such as $\tau^u$ in Eq. 8 does not coincide with $\tau_u$ in the previous text.
4.	It seems that the method proposed in this paper lacks comparison with previous works that also unified modeling aleatoric uncertainty and epistemic uncertainty, such as [1] use dirichlet distribution to model aleatoric uncertainty and epistemic uncertainty uniformly.
5.	Is it really reasonable to discard 5% of the validation set data of IID as OOD data? In essence, it seems that this 5% data is still ID data. This is confusing for me. Could you please give more explaination for me?

[1] Malinin, Andrey, and Mark Gales. "Predictive uncertainty estimation via prior networks." Advances in neural information processing systems 31 (2018).

**Questions:**

Please refer to weakness.

---

> ### Author Response · Authors · 2023-11-14
>
> Thank you  for your feedback!
>
> To address your questions:
>
> * Thank you for the reference, we will definitely discuss this. However, while the mentioned reference requires re-training prior networks from scratch, U2C can be applied with any frozen pre-trained model (torchvision, huggingface…), with no access to the original training data, and with very modest compute overhead (mostly to featurize the validation set).
>
> * The 5% of IID validation data that we select as “looking most like OOD” are those points that have the highest epistemic uncertainty, as measured by $u(x)$. In fact, Reject-or-Classify also gives up on these same 5% tail of examples.
>
> * Thank you for pointing out the typos and writing issues. You are correct that if P_in has more mass in C but little in B, then U2C outperforms RC. We will fix this, add clarifying pseudocode, and fix the other typos ($\tau_u$ to $\tau^u$).

---

### Official Review · Reviewer_WQhb · 2023-11-01

**Soundness:** 1 poor
**Presentation:** 1 poor
**Contribution:** 2 fair
**Rating:** 3
**Confidence:** 4

**Summary:**

A method is proposed to produce scores over the classes in a multiclass classification problem, as well as a score indicating that the sample is out-of-distribution. These scores sum to one. It is claimed that they are well-calibrated in-sample and have good out-of-sample detection error. It is suggested that in-sample calibration quantifies aleatoric uncertainty, and out-of-sample detection quantifies epistemic uncertainty.

**Strengths:**

The problem of separating aleatoric and epistemic uncertainty is interesting. The proposed method attacks one aspect of the aleatoric-epistemic uncertainty problem by considering the calibration and out-of-sample detection problem in multiclass classification jointly.

**Weaknesses:**

A lot of the paper has a philosophical feeling to it, involving comments on etymology, sources of randomness, and vague correct-sounding statements that are presented without citation. Sometimes, these statements are patently incorrect. Examples follow:
- Eq. (2) is an incorrect definition of ECE. ECE does not have an inner expectation over p \sim [0,1]. It is also strange (although technically correct) to write the outer expectation over (x, y) \sim P -- the outer expectation is only over x \sim P_X. Please see one of the references you have cited for the right measure-theoretic definition.
- The first sentence of page 4 is, "In practice, a classifier models aleatoric uncertainty if it is well calibrated (Guo et al., 2017; Wang et al., 2021), namely it satisfies \pi_f (x) ≈ P(Y = y | X = x) for all examples (x, y)."
Neither of the papers claim that calibration models aleatoric uncertainty, or that the learnt \pi_f is close to the true regression function, so this is an unsubstantiated opinion of the authors. Also, it is impossible to learn the true regression function, and none of these methods do it (or claim to).
- Section 3.1: "In machine learning research, aleatoric uncertainty arises due to irreducible sources of randomness in the process of labeling data." This is a strange statement that also falsely implies that all source of aleatoric uncertainty is annotator noise.
- Section 3.2: "Epistemic uncertainty is an unsupervised quantity, and as such it is more challenging to estimate than its supervised counterpart, aleatoric uncertainty." (no citation is provided)

Some other writing issues:
- The function u(x), central to the method of the paper, is mentioned inline as part of examples which were listed only for the sake of completeness (in the authors' words).
- There are two lemmas but no theorem.
- Changing notation: from \tau_u to \tau^u
- What is an example of a "non-linear" \tilde{\tau}_u in eq. (7)? This is so central to the method that it should be transparent at first reading.

The theoretical results are not noteworthy, and are better presented as an illustrative synthetic example involving some computation. Lemma 5.1 is a decomposition of errors. Lemma 5.2 verifies that RC does not do well on the given example.

I had some issues understanding the method, but I believe it essentially boils down to relabeling some percentage of the validation dataset as out-of-sample, then learning some model to predict those out-of-sample points. Then a softmax is applied in the end to make sure things sum to one. The method makes sense but I feel lacks the level of novelty expected from ICLR.

**Questions:**

- Eq. (4): Why is this a good way to separate aleatoric and epistemic uncertainty? Is there a specific paper that has justified the use of these?
- What is the purpose of including etymological remarks when introducing aleatoric and epistemic uncertainty?
- What is the "non-linear" \tilde{\tau}_u in eq. (7)?

---

> ### Author Response · Authors · 2023-11-14
>
> Thank you for your feedback!
>
> To address your main concerns:
>
> * For better clarity, we will redefine ECE as: $E_{\hat{P}} [ | \Pr(\hat{Y} = y | \hat{P} = p) - p |]$, as in [Guo et al, 2017].
>
> * You are correct: the softmax probabilities of a neural network measure aleatoric uncertainty only under the assumption that the network approaches the correct regression function $p(y|x)$. We will clarify this in the paper.
>
> * The non-linear $\tau^u$ we use in our experiments is a ReLU MLP of sizes 1-64-1. We will clarify this early on.
>
> Thank you for pointing out the typos and writing issues. We will fix the typos and address the remaining as follows:
>
> * We will clarify that there are other sources of aleatoric uncertainty other than annotator noise, and the difficulties involved in epistemic uncertainty estimation as those related to estimating high-dimensional supports.
> * We will relabel our lemmas as propositions.
> * The taxonomy of aleatoric and epistemic uncertainty was introduced in [https://arxiv.org/abs/1703.04977]. For a Bayesian decomposition, see [https://arxiv.org/abs/1710.07283]. We will clarify and add the references.

---

### Official Review · Reviewer_Pdw3 · 2023-11-01

**Soundness:** 3 good
**Presentation:** 4 excellent
**Contribution:** 3 good
**Rating:** 5
**Confidence:** 4

**Summary:**

The paper proposes a new way of combining uncertainties. Given a calibrated network f_\tau and an epistemic uncertainty estimator u, instead of doing reject-or-classify, the paper suggests to do the following: (1) use a new validation set {x_i, y_i} and relabel 5% of its data---that has the worst uncertainty scores u(x_i)---with a new label c+1 to obtain {x_i, y_i'}; (2) train a calibration function \tau_u that serves as the logit for the class c+1 using cross entropy loss between (f_\tau(x_i), \tau_u(x_i)) and one-hot(y_i'). Empirically, with different ucertainty estimators u, the proposed approach was able to improve the performance of the model in a couple of settings.


---- After Rebuttal ------
While there are some merits in the proposed unifying framework for uncertainty estimation and the authors have tried to address some of my questions, I discussed with other reviewers and shared similar concerns as them, hence I am adjusting my score accordingly.

**Strengths:**

The paper is very well written, with a clear overview of the field and an explanation of why reject-or-classify is not ideal. The proposed approach is simple (in a good sense) and well-motivated. The simplicity makes it applicable to all classification settings. The experimental results show the efficacy of the methods in a variety of settings.

**Weaknesses:**

The paper doesn't have major weaknesses to me, though I do have a couple of questions for the authors to answer listed below.

**Questions:**

- Does \tau^u need to be calibrated in some sense? For example, is it possible that adding \tau^u as the logits for class c+1 will mess up with the calibrated logits for the classes 1~c?
- The choice of u and f_\tau. Does the approach have some requirement on u? For example, given a very "u", the data chosen to be labeled as c+1 can be wildly wrong. Is there a way to choose u itself? If so, can it be integrated into this unified approach?
- Should \tau^u(u(x)) in Eq.8 be \tau_u(x)? Or maybe the objective (7) should be changed to \tau_u(u(x_i^va))?

---

> ### Author Response · Authors · 2023-11-14
> **Answers to your questions**
>
> Thank you for your feedback!
>
> To answer your questions,
>
> * This is indeed the case:  $\tau^u$ is jointly learned with the temperature parameter of the first C logits, by minimizing the cross entropy across a validation set, where 5% of examples have been relabeled as OOD (class C+1).
> * You are correct: both RC and U2C will model uncertainty poorly when given the wrong $u(x)$. In this work, we have considered $u(x)$ as given, and chosen from one of the many state-of-the-art solutions previously developed in the literature, such as k-nn or Mahalonobis distances.
> * We apologize for the typo; the term involving $\tilde{\tau}$ in equation (7) should read $\tilde{\tau}(u(x_i^{\text{va}})$.

---

### Meta-Review · Area_Chair_pzGY · 2023-12-05

**Metareview:**

This paper presents a method, referred to as “unified uncertainty calibration” (U2C), which is a holistic framework to combine both aleatoric and episdemic uncertainties when the calibration is performed. The method is extremely simple and the approach is reasonable. However, there are a few serious concerns raised by reviewers. Of particular concerns are unclarities in the writing, lack of theoretical justification, and some misleading claims. Details can be found in reviewers’ comments. During the AC-reviewer discussion period, after the author response was submitted, most of reviewers feel that the paper is not ready for being published on ICLR in its current version. Therefore, the paper is not recommended for acceptance in its current form. I hope authors found the review comments informative and can improve their paper by addressing these carefully in future submissions.

**Justification For Why Not Higher Score:**

Needs to improve the clarity in the writing and add some theoretical justication.

**Justification For Why Not Lower Score:**

N/A

---

### Decision · Program_Chairs · 2024-01-16

Reject